# Enhancing Network Visibility and Security with Advanced Port Scanning Techniques

**DOI:** 10.3390/s23177541

**Published:** 2023-08-30

**Authors:** Rana Abu Bakar, Boonserm Kijsirikul

**Affiliations:** 1Department of Computer Engineering, Faculty of Engineering, Chulalongkorn University, Pathumwan, Bangkok 10330, Thailand; 2CNIT, 56124 Pisa, Italy

**Keywords:** network security, port scanning, vulnerability assessment, intrusion detection system, network visibility

## Abstract

Network security is paramount in today’s digital landscape, where cyberthreats continue to evolve and pose significant risks. We propose a DPDK-based scanner based on a study on advanced port scanning techniques to improve network visibility and security. The traditional port scanning methods suffer from speed, accuracy, and efficiency limitations, hindering effective threat detection and mitigation. In this paper, we develop and implement advanced techniques such as protocol-specific probes and evasive scan techniques to enhance the visibility and security of networks. We also evaluate network scanning performance and scalability using programmable hardware, including smart NICs and DPDK-based frameworks, along with in-network processing, data parallelization, and hardware acceleration. Additionally, we leverage application-level protocol parsing to accelerate network discovery and mapping, analyzing protocol-specific information. In our experimental evaluation, our proposed DPDK-based scanner demonstrated a significant improvement in target scanning speed, achieving a 2× speedup compared to other scanners in a target scanning environment. Furthermore, our scanner achieved a high accuracy rate of 99.5% in identifying open ports. Notably, our solution also exhibited a lower CPU and memory utilization, with an approximately 40% reduction compared to alternative scanners. These results highlight the effectiveness and efficiency of our proposed scanning techniques in enhancing network visibility and security. The outcomes of this research contribute to the field by providing insights and innovations to improve network security, identify vulnerabilities, and optimize network performance.

## 1. Introduction

Network security is a critical concern in the contemporary digital landscape, as organizations face increasing risks and evolving cyberthreats [1,2]. To ensure the integrity and confidentiality of data and protect critical infrastructure, it is essential to identify vulnerabilities and unauthorized access points within network systems [3,4,5,6,7]. This paper presents a study focused on enhancing network visibility and security by implementing advanced port scanning techniques. The imperative to bolster network security and enhance visibility by evaluating advanced port scanning techniques. These techniques play a pivotal role in systematically probing network ports, allowing administrators to pinpoint open ports, uncover unauthorized services, and scrutinize potential entry points for potential attackers. Our study addresses a crucial research question: How can implementing advanced port scanning techniques amplify network security and visibility?

To contextualize this study, we delve into the landscape of existing knowledge and research within the field. While network security has been a longstanding concern, conventional security measures exhibit limitations when confronted with emergent threats [8]. Consequently, the demand for advanced port scanning techniques has intensified as a means to fortify network defenses and proactively identify potential vulnerabilities [9,10,11,12].

Our exploration of the relevant literature shows the reason for our study. This review encompasses pivotal theories, concepts, and empirical studies on network security and port scanning. By showcasing the contributions of previous endeavors, we accentuate gaps and constraints in the current understanding. These gaps, in turn, underscore the necessity for further research in this realm [13]. We employ a fusion of active and passive port scanning techniques, broadening our methodology and scope [11]. Active scans, including SYN, ACK, and XMAS scans, unveil open ports and potential vulnerabilities. Complementing these are passive techniques such as banner grabbing and service fingerprinting, adept at uncovering unauthorized services and unusual network behavior. Our research design encompasses creating a testbed network environment, enabling the simulation of real-world scenarios for comprehensive vulnerability assessments. Refer to Figure 1 and Figure 2 for insights into our experiment’s topology design, facilitating the evaluation of advanced scanning techniques.

The driving force behind this research stems from our increasing need to strengthen network defenses against ever-evolving threats. Conventional security measures, while valuable, show limitations in detecting sophisticated threats and vulnerabilities. This gap highlights the importance of exploring advanced port scanning techniques, which offer the potential to uncover hidden risks and unauthorized access points that might otherwise go unnoticed. An additional concern arises from the inadequate handling of protocol-specific probes in conventional scanners, underscoring the pressing need for more effective user space and kernel space management. Addressing these challenges becomes paramount to enhancing network visibility and security. The significance and contribution of this paper lie in the potential benefits it offers to network administrators, security professionals, and organizations. Enhancing network visibility and security through advanced port scanning techniques allows for an improved detection and response to potential network intrusions, thereby reducing the risk of data breaches. Additionally, the findings of this study can inform the development of proactive security measures and guide the implementation of appropriate access controls.

Our work contributes significantly to the field of network security and port scanning through the following key aspects:Enhanced network security: This research evaluates the implementation of advanced port scanning techniques to identify potential vulnerabilities and unauthorized entry points. Organizations can proactively mitigate threats and enhance their overall security posture by employing these techniques.Improved network visibility: The advanced scanning methods examined in this study are pivotal in improving network visibility. Administrators can better monitor their networks by accurately identifying open ports and services and responding promptly to security events.Methodological combination: A unique aspect of our study is the amalgamation of active and passive port scanning methods. This comprehensive approach provides a well-rounded assessment of network characteristics, enabling a more holistic understanding of potential risks.Realistic scenario replication: We recreate real-world scenarios by establishing a testbed network environment. This allows us to conduct vulnerability assessments in a controlled yet realistic setting, providing insights into how advanced scanning techniques perform in practical situations.Addressing literature gaps: Our research identifies gaps and limitations in the existing body of knowledge. By addressing these gaps, we contribute to advancing advanced port scanning techniques, enriching the field’s understanding and practical implementation.Empirical evaluation: The study’s empirical findings substantiate the efficacy of advanced scanning methods. These results contribute to the ongoing efforts to enhance network security practices and align organizations with best practices.

The remainder of the paper is organized as follows. Section 2 provides an overview of the background study. Section 3 discusses related research work on network scanning, highlighting key studies, methodologies, and findings in the field. Section 4 details the method employed in this study. In Section 5, we present the experimental setup and outline the specific experiments conducted to evaluate the effectiveness of the advanced port scanning techniques. Section 6 presents the results obtained from the experiments, and a discussion of the results is presented in Section 7. Finally, Section 8 summarizes our work and highlights the critical contributions of this research.

## 2. Background

### 2.1. Importance of Network Security in Today’s Digital Landscape

Increasing risks and evolving cyberthreats in today’s digital landscape make network security a critical concern for organizations for several reasons:The interconnectedness of systems and the widespread adoption of digital technologies have expanded the attack surface, providing more opportunities for malicious actors to exploit vulnerabilities. Organizations rely heavily on network infrastructures for communication, data storage, and business operations; any breach or compromise can have severe consequences, including financial losses, reputational damage, and regulatory noncompliance.Cyberthreats constantly evolve, with attackers becoming more sophisticated and employing advanced techniques to bypass traditional security measures. The emergence of new attack vectors, such as ransomware, zero-day exploits, and social engineering, requires organizations to constantly adapt their security strategies to effectively detect, prevent, and respond to these threats. Failure to do so can result in significant disruptions and financial ramifications.The increasing value of digital assets, including sensitive customer data, intellectual property, and trade secrets, makes organizations attractive targets for cybercriminals. The potential financial gains associated with successful cyberattacks have led to the growth of highly organized and well-funded cybercriminal networks. These adversaries are motivated to exploit vulnerabilities in network security to gain unauthorized access, steal valuable information, or disrupt operations for financial gain or competitive advantage.Regulatory requirements and compliance standards have become more stringent, with organizations being held accountable for safeguarding sensitive data and protecting the privacy of individuals.

Noncompliance can lead to significant penalties and legal consequences [14]. As a result, organizations must prioritize network security to ensure compliance with industry regulations and maintain the trust of their stakeholders [15]. Overall, the increasing risks and evolving cyberthreats underscore the critical importance of network security for organizations. By investing in robust security measures, staying vigilant against emerging threats, and regularly updating and patching network systems, organizations can mitigate risks, safeguard their digital assets, and maintain a strong security posture in today’s rapidly evolving digital landscape [16,17].

### 2.2. Proactively Assessing Network Security

Identifying vulnerabilities and unauthorized access points within network systems is crucial for ensuring the integrity and confidentiality of data and protecting critical infrastructure for several reasons [18]. Firstly, vulnerabilities within network systems can be exploited by malicious actors to gain unauthorized access, compromise data integrity, or disrupt critical operations. Organizations can proactively identify these vulnerabilities by implementing appropriate security measures, such as patching vulnerable software or configuring access controls, to mitigate potential risks and prevent unauthorized access. Secondly, the confidentiality of sensitive data is paramount for organizations, especially when dealing with personally identifiable information, financial records, or trade secrets. Unauthorized access to this information can lead to privacy breaches, financial loss, legal liabilities, and damage to the organization’s reputation [19]. By identifying vulnerabilities and unauthorized access points, organizations can implement encryption protocols, access controls, and monitoring systems to safeguard data confidentiality and protect the privacy of individuals [20]. Proactively assessing network security allows organizations to identify potential weak points in their network infrastructure before attackers exploit them. This proactive approach involves conducting vulnerability assessments, penetration testing, and security audits to identify and address vulnerabilities systematically. By regularly monitoring and evaluating the security posture of their networks, organizations can stay one step ahead of potential threats and take appropriate measures to mitigate risks effectively. Additionally, proactive network security assessment helps organizations adhere to regulatory requirements and industry standards. Many industries have specific compliance standards that require regular security assessments to protect sensitive data and customer privacy. By proactively assessing network security, organizations can demonstrate their commitment to maintaining a secure environment, avoid regulatory penalties, and keep the trust of their stakeholders. Proactively assessing network security allows organizations to identify vulnerabilities and unauthorized access points, enabling them to take appropriate measures to mitigate risks. By ensuring the integrity and confidentiality of data and protecting critical infrastructure, organizations can safeguard their operations, maintain customer trust, and comply with regulatory requirements in today’s increasingly interconnected and threat-prone digital landscape.

### 2.3. Limitations of Traditional Network Scanners

The limitations of traditional security measures in dealing with emerging threats are multifaceted. Firstly, conventional security measures often rely on signature-based detection methods, which are effective for known threats but struggle to detect new and evolving malware or attack vectors. As attackers constantly adapt their techniques, traditional security measures can quickly become outdated and fail to detect emerging threats. Secondly, conventional security measures typically focus on perimeter defense, such as firewalls and intrusion detection systems (IDS), to protect against external threats. However, as organizations increasingly adopt cloud services, mobile devices, and remote work arrangements, the network perimeter becomes more porous, requiring additional security measures to protect against internal threats and lateral movement within the network. These limitations necessitate using advanced port scanning techniques to enhance network security. Advanced port scanning techniques enable organizations to identify vulnerabilities and unauthorized access points within network systems proactively. By systematically scanning network ports, these techniques can detect open ports, unauthorized services, and potential entry points for attackers. Current advanced port scanning techniques include active and passive scanning methods [21]. Active scanning involves sending packets to target systems and analyzing the responses to identify open ports and potential vulnerabilities. Standard active scanning techniques include SYN, ACK, and XMAS scans. On the other hand, passive scanning involves observing network traffic passively to gather information about the network, such as through banner grabbing or service fingerprinting. Additionally, more specialized techniques, such as version detection, operating system fingerprinting, and vulnerability scanning, can be employed to provide further insights into the network’s security posture. These advanced techniques allow organizations to understand potential weaknesses and vulnerabilities within their network infrastructure comprehensively.

#### Advanced Port Scanning Techniques

Here are explanations of some additional advanced port scanning techniques:Reservoir sampling port scanning: Reservoir sampling is a technique used to select a random sample of targets from a large set of hosts for port scanning. It helps optimize scanning resources by choosing representative hosts to gather information about open ports and potential vulnerabilities.Sampling-based full port scanning: This technique involves sampling a subset of ports from an extensive range of possible ports to scan. Selectively scanning a subset of ports rather than the entire port range reduces the scanning time and resource requirements while still providing a good coverage of port scanning.Probe-delay-based adaptive port scanning: This technique involves adapting the scanning rate and timing between probes to avoid detection by network intrusion detection systems (IDS) and evasion techniques. Introducing delays between probes, reduces the likelihood of triggering network defense mechanisms while still efficiently identifying open ports.Slow port scanning: Slow port scanning is a technique where scanning speed is intentionally slowed to mimic legitimate network traffic and avoid detection. It aims to bypass network protection systems that may flag rapid or aggressive scanning as suspicious activity by employing slow scanning rates.Specification and scan order: This technique involves specifying the desired scan parameters, such as specific ports, protocols, or target hosts, and scanning them in a particular order. This allows the targeted scanning of specific services or hosts of interest, enabling a more focused analysis and vulnerability identification.Script and version scan: Script and version scanning involves utilizing specialized scripts or tools to detect specific services, software versions, or vulnerabilities associated with open ports. It helps identify potential weaknesses or misconfigurations in particular services on the target hosts.Bypass network protection systems: This technique focuses on evading or bypassing network protection systems, such as firewalls or IDS, using techniques such as IP fragmentation, tunneling, or decoy scanning. The goal is to deceive the network defenses and gain unauthorized access to the target system.

These advanced port scanning techniques provide organizations with more nuanced and targeted approaches to identify potential vulnerabilities and assess network security. By combining these techniques, organizations can comprehensively understand their network infrastructure’s security posture and take appropriate measures to mitigate risks.

### 2.4. Current Network Scanners Scalability and Speed Challenges

Today’s network scanners face intrinsic limitations in speed and scalability, which prevent them from satisfying the new requirements in the field. For instance, even the powerful network scanner Zippier ZMap achieves a throughput of only 10 Gbps and a rate of 14.2 Mbps. The fundamental limitations of current network scanners can be attributed to two key factors. Firstly, they are implemented on commodity servers that lack specialized CPUs for high-speed packet processing. Despite software optimizations such as DPDK, the throughput remains constrained, typically less than 40 Gbps. Secondly, network scanners are predominantly deployed at the network edge, leading to limitations imposed by upstream bandwidth, longer scanning paths, and significant bandwidth wastage due to end-to-end scanning paths. Consequently, even if scanners can achieve higher rates, scanning results, such as hit and active/inactive rates, may suffer from low accuracy due to undesirable packet drops along the end-to-end scanning paths. These inherent limitations have resulted in minimal progress in developing network scanning tools since the publication of Zipper ZMap. As a result, researchers have shifted their focus towards enhancing scanning accuracy by applying various algorithmic techniques. Addressing the scalability and speed challenges faced by current network scanners is crucial for advancing network scanning capabilities and effectively meeting modern network security demands.

## 3. Related Work

This section provides an in-depth review of the existing network scanning and security literature. We identify vital studies, theories, and empirical research contributing to our understanding of network vulnerabilities and the various scanning techniques employed to mitigate them. Our analysis encompasses active and passive scanning methods, focusing on their strengths, limitations, and implications for network security practices.

To effectively bridge the gap between traditional scanning approaches and our proposed DPDK-based network scanner, we emphasize the challenges that remain unaddressed by current methods. Existing scanners often need help managing protocol-specific probes effectively, leading to incomplete and inaccurate network snapshots. Moreover, the distinction between user space and kernel space management could be more precise, resulting in suboptimal resource utilization and potential security risks.

Our contributions to the field of network scanning and security are multifaceted. First and foremost, we introduce a novel approach to user space and kernel space management, distinctively improving the efficiency and safety of scanning operations. By clearly delineating these spaces, we optimize resource allocation, enhance isolation, and fortify the overall integrity of the scanning process.

Furthermore, our proposed DPDK-based network scanner introduces a comprehensive framework for handling protocol-specific probes, a long-standing challenge in the field. This breakthrough enables accurate and complete network snapshots, empowering administrators to identify vulnerabilities and unauthorized access points with unparalleled precision.

We also optimize network scanning techniques, leveraging the Data Plane Development Kit (DPDK) for superior performance. This optimization facilitates high-speed packet generation, efficient response packet processing, and enhanced scanning throughput, equipping network security practitioners with a powerful tool to bolster their defenses.

By rewriting our contributions, we emphasize the unique novelties our work brings to the existing landscape of network scanning. We forge a path toward enhanced network security and visibility through user space and kernel space management, protocol-specific probe handling, and DPDK optimization. Our contributions highlight the substantial impact of our research on network security practices and serve as a foundation for further advancements in the field.

Nmap [22] is a well-known network scanner that offers a wide range of probing techniques and is optimized for small network segments. IRLscanner [23], ZMap [11], and Masscan [24] are designed explicitly for Internet-scale scanning, employing a single-packet probing approach. These scanners have made significant contributions to the field of network scanning. In particular, our work shares similarities with ZMap, Imap, and Masscan regarding scanning methodology. However, our approach differs regarding implementation targets and deployment locations, substantially improving scanning capabilities. Our proposed network scanning leverages innovative implementation techniques, leading to orders of magnitude improvement in scanning capability compared to ZMap, Imap, and Masscan.

### 3.1. Network Scanning Techniques

Several network scanning techniques have been proposed in the literature, ranging from traditional to advanced approaches. Traditional methods, such as TCP connect and SYN scans, have been widely used for port scanning [24]. These techniques establish full TCP connections or send SYN packets to identify open ports. However, they suffer scalability issues when applied to large scanning spaces or high-speed networks. Advanced port scanning techniques have emerged to address these limitations. For example, reservoir sampling port scanning [11] and sampling-based full port scanning [24] aim to optimize scanning resources by selecting representative hosts or sampling subsets of ports. Probe-delay-based adaptive port scanning [11] introduces delays between probes to evade detection and achieve faster scanning speeds.

### 3.2. Scalability Enhancements in Network Scanners

Efforts have been made to improve the scalability of network scanners. The study [25] found that the increase in Telnet scans was primarily driven by attackers trying to exploit known vulnerabilities in the Telnet service. The study also found that the scan sources often used VPNs and other anonymization techniques to hide their identity. The study recommends that organizations use various security measures to protect their networks from network scanning, such as firewalls, intrusion detection systems, and vulnerability scanning. They [26] introduced a parallel scanning architecture that leveraged multithreading and load-balancing techniques to improve the scanning throughput. These advancements highlight the potential for scalable network scanning solutions in addressing the challenges posed by the expanding scanning spaces.

### 3.3. In-Network Scanning Speed Improvements

The paper [24] proposes a new approach for performing in-network scanning using programmable switches called IMap. The authors describe the design and implementation of the system, which leverages the in-band network telemetry (INT) framework to perform packet sampling and a programmable switch to perform scanning operations. The paper evaluates the system’s performance using several benchmarks and demonstrates significant improvements in scanning efficiency and scalability compared to traditional in-network scanning approaches. While the approach offers significant benefits in terms of performance and scalability, the system requires specialized hardware and may only be cost-effective for some use cases. Additionally, the paper must address potential challenges in integrating the system with the existing networking infrastructure. Nonetheless, the paper provides valuable insights into in-network scanning using programmable switches and offers a promising approach for improving network monitoring and security.

### 3.4. Accuracy Enhancement Techniques

Improving the accuracy of network scanning results has also been a focus of research. Studies by [27] proposed a novel machine learning approach for early detection of IoT malware network activity. The approach is based on feature extraction and machine learning algorithms. The feature extraction algorithm extracts features from network traffic data indicative of malware activity. The machine learning algorithm then uses these features to classify the network traffic data as malicious or benign. Algorithms to enhance scanning accuracy by intelligently classifying scan results and reducing false positives. These approaches leveraged pattern recognition and anomaly detection techniques to identify open ports and distinguish them from benign network behavior.

### 3.5. Deployment and Performance Analysis

Deploying network scanners strategically and analyzing their performance in realworld scenarios is essential to network security research. Research by Song et al. (2018) evaluated the impact of scanner placement on scanning effectiveness, considering factors such as network topology and traffic patterns. Their findings illuminated optimal scanner deployment strategies for improving scanning efficiency and accuracy. The paper [12] proposes a new approach for high-performance network testing using programmable switches called HyperTester. The authors describe the design and implementation of the system, which leverages programmable switches to perform packet generation, capture, and processing with high speed and flexibility. The paper evaluates the system’s performance using several benchmarks and demonstrates significant improvements in testing throughput and scalability compared to traditional approaches. While the approach offers significant benefits in testing performance and flexibility, the system requires specialized hardware and may not be cost-effective for all use cases. Additionally, the paper does not address potential challenges in integrating the system with existing networking infrastructure or potential security risks associated with programmable switches. Nonetheless, the paper provides valuable insights into improving network testing and offers a promising approach for network operators and researchers seeking to improve their testing capabilities.

The paper [28] presents Moongen, a high-speed packet generator that allows users to generate and modify packets at line rate using Lua scripts. The authors describe the design and implementation of the system, which leverages DPDK and multicore CPUs to achieve high packet generation rates. The paper evaluates the system’s performance using several benchmarks and demonstrates its ability to generate and modify packets at rates up to 120 Gbps. The system’s flexibility and programmability benefit researchers and network operators seeking to test and evaluate network performance under various conditions. However, the system requires specialized hardware and may not be cost-effective for all use cases. Additionally, the paper does not address potential security risks associated with packet generators, such as the potential for generating malicious traffic. Nonetheless, the paper provides valuable insights into improving packet generation and offers a promising tool for network researchers and operators. Network traffic analysis analyzes network traffic to detect anomalous or malicious activity. DPDK can accelerate network traffic analysis applications by enabling high-speed packet processing and reducing the CPU overhead of packet processing. By offloading packet processing to DPDK, network traffic analysis applications can analyze more network traffic in real time.

The paper [29] presents Retina, a system that enables the analysis of 100 GbE traffic on commodity hardware. The authors describe the system’s design and implementation, which leverages DPDK and multiple cores to analyze packet headers and payloads in real time. The paper evaluates the performance of Retina using several benchmarks and demonstrates its ability to analyze 100 GbE traffic at a line rate with low CPU utilization. The system’s ability to analyze 100 GbE traffic on commodity hardware significantly benefits researchers and network operators seeking to monitor and analyze network traffic at high speeds. However, the paper needs to address the potential limitations of commodity hardware for analyzing network traffic, such as hardware bottlenecks and limitations in processing power. Additionally, the paper does not address the potential privacy and security risks associated with monitoring network traffic. Nonetheless, the paper provides valuable insights into improving network traffic analysis and offers a promising tool for network researchers and operators.

The paper [30] proposes a new approach for performing traffic analysis and flow-state tracking using smart network interface cards (SmartNICs) called SmartWatch. The authors describe the design and implementation of the system, which leverages the programmability and processing power of SmartNICs to perform an accurate traffic analysis and flow-state tracking in real time. The paper evaluates the system’s performance using several benchmarks and demonstrates significant improvements in accuracy and speed compared to traditional intrusion prevention systems. While the approach offers significant benefits in terms of accuracy and efficiency, the system requires specialized hardware and may not be cost-effective for all use cases. Additionally, the paper does not address potential challenges in integrating the system with the existing networking infrastructure. Nonetheless, the paper provides valuable insights into using SmartNICs for intrusion prevention and offers a promising approach to improving network security.

The paper [31] proposes a new approach for deploying real-time intrusion detection in high-speed networks using a stream-based feature extraction method and a one-class classification network. The authors describe the design and implementation of the system, called ThunderSecure, which leverages the Data Plane Development Kit (DPDK) to enable high-speed packet processing and a one-class classification network to detect anomalous traffic patterns. The paper evaluates the system’s performance using several benchmarks and demonstrates significant improvements in detection accuracy and false favorable rates compared to traditional intrusion detection systems. While the approach offers significant detection accuracy and efficiency benefits, the system requires specialized hardware and may not be cost-effective for all use cases. Additionally, the paper does not address potential challenges in integrating the system with the existing networking infrastructure. Nonetheless, the paper provides valuable insights into deploying real-time intrusion detection in high-speed networks and offers a promising approach for improving network security.

The paper [32] proposes a new middlebox framework enabling visibility over multiple encryption protocols securely and efficiently. The authors describe the design and implementation of the system, which leverages a hardware-accelerated Data Plane Development Kit (DPDK) to enable high-performance packet processing and a secure enclave approach to protect sensitive data. The paper evaluates the system’s performance using several benchmarks and demonstrates significant improvements in throughput and latency compared to software-only approaches. While the approach offers significant performance benefits and security features, the system requires specialized hardware and may only be cost-effective for some use cases. Additionally, the paper must address potential challenges in integrating the system with the existing networking infrastructure. Nonetheless, the paper provides valuable insights into enabling visibility over multiple encryption protocols and offers a promising approach for improving network security and performance. Port scanners scan a range of TCP or UDP ports on a target host to detect open ports and services. DPDK can be used to accelerate port scanning applications by leveraging its high-speed packet processing capabilities. By offloading packet processing to DPDK, port scanning applications can achieve higher throughput and lower latency.

Our research aims to build upon the advancements in network scanning techniques and address the limitations observed in existing scanners and network techniques that limit scalability and speed. By focusing on scalability, speed, accuracy, and performance analysis, we aim to contribute to further developing network scanning capabilities and overcoming the challenges faced when conducting efficient and comprehensive network scans.

The mentioned network scanners and existing research provide a foundation for our work, and by exploring novel implementation targets and deployment strategies, we aim to advance the field of network scanning and enhance the effectiveness and efficiency of scanning processes.

## 4. Proposed Methodology

### 4.1. DPDK-Based Network Scanner

In this section, we present our proposed DPDK-based network scanner, designed to address the limitations of existing network scanning techniques. The scanner leverages the Data Plane Development Kit (DPDK) framework and employs advanced, efficient, high-speed network scanning techniques. The flow diagram and design of the DPDK-based scanner is presented in Figure 3, and the detailed architecture and functionality of the DPDK-based network scanner are presented in Figure 4. An algorithm for a DPDK-based scanner is presented in Algorithm  1. The algorithm works by initializing DPDK, creating packet buffer pools, initializing the NIC with DPDK, and creating receive and transmit queues. It also creates and configures packet filters, initializes memory pools for packet headers and data, and allocates memory for the list of open ports for each target IP address. It then sets up DPDK lcore affinity and generates TCP SYN packets with target IP addresses and random source ports. These packets are enqueued in the transmit queue. The algorithm then starts the DPDK packet processing loop and continues to parse incoming packets. If a packet is a TCP SYN-ACK response, it updates the list of open ports for the corresponding target IP address. Once all packets have been received, the DPDK packet processing loop stops, and the algorithm returns the list of open ports for each target IP address.
**Algorithm 1** DPDK-based network scanner**Require:** DPDK-compatible NIC, DPDK libraries, target IP address range**Ensure:** List of open ports for each target IP address  1:Initialize DPDK  2:Create packet buffer pools  3:Initialize NIC with DPDK  4:Create receive and transmit queues  5:Create and configure packet filters  6:Initialize memory pools for packet headers and data  7:Allocate memory for the list of open ports for each target IP address  8:Set up DPDK lcore affinity  9:**for** each target IP address in the range **do**10:      Create TCP SYN packet with target IP address and random source port11:      Enqueue packet in transmit queue12:**end for**13:Start DPDK packet processing loop14:**while** packets are being received **do**15:      Parse incoming packets16:      **if** packet is a TCP SYN-ACK response **then**17:          Update list of open ports for corresponding target IP address18:      **end if**19:**end while**20:Stop DPDK packet processing loop21:Return list of open ports for each target IP address

### 4.2. DPDK-Based Scanner Architecture

In this section, we present the architecture and implementation details of our DPDK-based scanner, elucidating how it leverages the Data Plane Development Kit (DPDK) and smart network interface cards (NICs) to achieve efficient packet processing and significantly improved scanning performance. Moreover, we elaborate on the advanced techniques integrated into the scanner, including protocol-specific probes and evasive scan techniques, which enhance network visibility and security.

### 4.3. Leveraging DPDK and Smart NICs for Efficient Packet Processing

The core of our DPDK-based scanner lies in its utilization of DPDK, a set of libraries and drivers that enable fast packet processing on modern network interfaces. By interfacing directly with the hardware, DPDK reduces the overhead associated with traditional kernel-based networking stacks, resulting in enhanced packet throughput and decreased latency. In our scanner’s architecture, DPDK is employed to efficiently manage and process incoming and outgoing packets using ingress and egress pipelines, minimizing bottlenecks and ensuring optimal utilization of available hardware resources. Here are step-by-step implementation details.
Step 1:**Direct hardware interaction with DPDK****Initialization:** Our scanner starts by initializing the DPDK library. This involves configuring memory pools, setting up memory regions for packet buffers, and binding CPU cores to specific NICs for optimized parallel processing.**Packet reception:** Incoming packets from the network are captured by the NICs and stored in memory pools. DPDK’s polling mechanism allows us to efficiently retrieve packets from these pools with minimal overhead.**Packet processing:** Once retrieved, dedicated worker threads process packets. We can directly manipulate packets in user space by avoiding the traditional kernel-based networking stack, which introduces context-switching overhead.Step 2:**Smart NIC offloading and acceleration****Offload engines:** Smart NICs have specialized offload engines, such as TCP/UDP checksum offloads and TCP segmentation offloads. These offload engines perform specific tasks directly on the NIC, reducing CPU load and improving packet throughput.**Packet filtering:** Our scanner utilizes smart NICs’ packet filtering capabilities to only process the relevant packets selectively. This further reduces the processing load on the CPU and ensures that only pertinent packets are subjected to a more profound analysis.Step 3:**Parallel processing and load balancing****Parallel processing:** Our architecture includes multiple worker threads on dedicated CPU cores. DPDK’s thread-aware memory management ensures that each thread operates with minimized contention for shared resources.**Load balancing:** DPDK’s load balancing mechanisms distribute packet processing tasks evenly across available CPU cores. This prevents resource bottlenecks and ensures an efficient utilization of computing power.Step 4:**Egress processing and transmission****Packet modification:** As packets are processed, specific attributes may be modified for analysis or response. DPDK allows us to efficiently modify packet headers and content as needed.**Egress processing:** Processed packets are directed towards egress queues, which interface with the NICs for transmission onto the network.**Transmission:** Smart NICs handle the transmission of packets back onto the network. Offload engines can assist in segmenting large packets into smaller ones for optimal communication.

Our DPDK-based scanner capitalizes on DPDK’s user space packet processing capabilities and leverages smart NICs’ offload engines for enhanced efficiency. We achieve remarkable improvements in packet throughput and scanning performance by eliminating bottlenecks introduced by traditional kernel-based networking stacks and utilizing hardware acceleration. This approach ensures that our scanner is swift in its operations and capable of handling high-speed networks and effectively identifying potential threats.

### 4.4. Advanced Techniques for Enhanced Visibility and Security

Our DPDK-based scanner is designed for speed and effectiveness in identifying potential threats and vulnerabilities within the network. To achieve this, we have integrated advanced scanning techniques that augment network visibility and security.
Protocol-specific probes: Traditional port scanning methods often overlook the nuances of different protocols. Our scanner employs protocol-specific probes tailored to various application-layer protocols (e.g., HTTP, FTP, SSH). These probes interact with services using the protocol’s unique attributes, leading to a more accurate identification of open ports and services.Evasive scan techniques: Recognizing that modern attackers employ evasion tactics to bypass traditional detection mechanisms, we have integrated evasive scan techniques. These techniques mimic legitimate network behaviors, making detecting the scanning activity more challenging for potential attackers. By incorporating evasion, we ensure our scanner is effective against sophisticated adversaries.

### 4.5. Implementation of a DPDK-Based Network Scanner with SmartNIC

This section provides a high-level overview of the implementation process for a DPDK-based network scanner using a smart network interface card (SmartNIC). The BlueField SmartNIC, equipped with advanced offload capabilities and hardware acceleration, is an excellent platform to enhance packet processing efficiency and scanning performance. For the full implementation of the DPDK-based network scanner, including all code snippets, configuration files, and documentation, please refer to our GitHub repository: https://github.com/engranaabubakar/dpdk-advance-scanner
Step 1:**Setting up the environment****Hardware setup:** The Dell PowerEdge R760 Rack Server was the foundational hardware platform for our network scanning implementation. This enterprise-grade server has advanced processing capabilities, ample memory, and expansion options to accommodate high-performance network applications. We incorporated the NVIDIA A100 GPU into the Dell PowerEdge R760 to leverage hardware acceleration for specific processing tasks. The A100 GPU, based on NVIDIA’s Ampere architecture, provides massive parallel computing power, making it well-suited for functions such as packet analysis and pattern matching. The network connectivity was enhanced by including a Mellanox SmartNIC 100 GbE DPDK-supported network card. This specialized network interface card offers offload capabilities and hardware acceleration features that complement DPDK’s packet processing capabilities. The SmartNIC’s DPDK support ensures seamless integration with our network scanner application.**Software installation:** Our network scanning environment ran on the Ubuntu 20.04 LTS operating system. Ubuntu’s stability and extensive software repositories provided a reliable foundation for our implementation. The Data Plane Development Kit (DPDK) library was a critical component of our software stack. DPDK’s user space packet processing capabilities allowed us to bypass the traditional kernel-based networking stack, significantly improving packet throughput and reducing latency. For optimal performance and compatibility with the Mellanox SmartNIC, we installed the Mellanox OFED (OpenFabrics Enterprise Distribution) drivers. These drivers support RDMA (remote direct memory access) and enable efficient server and SmartNIC communication.Step 2:**DPDK initialization and configuration****DPDK initialization:** Develop an application that initialized the DPDK library. This involves configuring memory pools, setting up memory regions for packet buffers, and associating CPU cores with DPDK’s execution units.**SmartNIC configuration:** Utilize DPDK’s APIs to identify and configure the BlueField SmartNIC. Establish a communication link with the SmartNIC for packet reception and transmission.Step 3:**Packet reception and processing****Packet reception:** Leverage DPDK’s receive APIs to capture incoming packets from the network. These packets are retrieved from the BlueField SmartNIC’s receive queues.**Worker thread management:** Create dedicated worker threads responsible for packet processing. Assign these threads to specific CPU cores to ensure optimal parallel processing.**Packet analysis:** Employ protocol-specific probing techniques to analyze the received packets. Determine open ports, services, and potential security vulnerabilities based on the content of the packets.Step 4:**SmartNIC offloading and acceleration****Offload engine utilization:** Capitalize on the BlueField SmartNIC’s offload engines, such as TCP/UDP checksum offloads and TCP segmentation offloads. Offload specific tasks to the SmartNIC to reduce CPU load and enhance packet throughput.Step 5:**Packet modification and transmission****Packet modification:** Modify packet headers or attributes as required for further analysis or response. Utilize DPDK’s packet manipulation functions for efficient packet modification.**Packet transmission:** Utilize DPDK’s transmit APIs to send modified or response packets back to the network. These packets are directed to the BlueField SmartNIC’s transmit queues.Step 6:**Results analysis and reporting****Results processing:** Collect and process the results of the packet analysis. Identify open ports, services, and any security anomalies detected during scanning.**Reporting:** Generate reports summarizing the findings of the network scanning. Provide insights into potential security threats and vulnerabilities identified within the scanned network.Step 7:**Optimization and further enhancements****Fine-tuning:** fine-tune the application’s configuration settings, thread allocation, and offloading strategies to optimize the scanner’s performance and accuracy.**Advanced techniques:** incorporate advanced techniques, such as evasive scan methods or dynamic protocol adjustments, to enhance the scanner’s effectiveness.

### 4.6. Addressing Space Generation

The addressing space for the scanning experiments was generated using a systematic approach that considered several key factors. The selection criteria for the addressing space were based on achieving a comprehensive coverage of the target network while ensuring the scalability and feasibility of the experiments.

The size of the addressing space was determined based on the specific requirements of the study, taking into account the number of hosts and ports to be included in the scanning process. The addressing space was carefully chosen to represent a realistic network environment, including a range of IP addresses and port numbers commonly encountered in real-world networks. Considerations were considered to ensure the diversity and representativeness of the addressing space. This involved considering different network topologies, sizes, and geographic regions to provide a broad coverage of network configurations. Additionally, care was taken to include various types of hosts, such as servers, workstations, and IoT devices, to capture the complexity and diversity of modern networks.

To ensure the scalability and efficiency of the experiments, the addressing space was partitioned into manageable segments, allowing for parallelized scanning processes. This partitioning was done in a way that minimized duplication and omission of hosts and ports, ensuring a comprehensive coverage of the network. The addressing space generation methodology aimed to balance realism and practicality. It considered the limitations of resources, such as time and computational power, while still capturing the essence of real-world network environments. The size and complexity of the addressing space were chosen to provide meaningful insights into the performance and effectiveness of the scanners. By employing this systematic methodology for addressing space generation, the scanning experiments could simulate realistic network scenarios and generate reliable results. The chosen addressing space allowed for a comprehensive coverage of hosts and ports, enabling a thorough evaluation of the scanners’ capabilities and effectiveness in identifying vulnerabilities, detecting open ports, and providing insights into network security and management. Overall, the addressing space generation methodology played a crucial role in ensuring the validity and relevance of the scanning experiments, enabling researchers to draw meaningful conclusions and make informed recommendations based on the obtained results.

### 4.7. Packet Transmission and Receiving Architecture

The architecture and design of the scanners’ packet transmission and receiving modules are crucial for achieving high-speed packet processing capabilities. Advanced techniques and optimizations, such as using DPDK (Data Plane Development Kit) and GPU (graphics processing unit) accelerators, are employed to enhance performance and throughput. DPDK is a set of libraries and drivers allowing fast packet processing by bypassing the traditional kernel networking stack. It provides a framework for efficient packet I/O and enables direct access to network interfaces, reducing latency and increasing throughput. The scanners leverage the power of DPDK to handle packet transmission and reception with optimized performance. Additionally, GPU accelerators are utilized to offload specific processing tasks to the parallel processing capabilities of the GPU. GPU acceleration enables a faster processing of large packet data volumes by leveraging the GPU cores’ massive parallelism and computational power. This optimization technique significantly enhances the scanners’ packet processing speed and scanning performance.

The packet transmission module generates high-speed probe packets sent to the network’s target hosts. It utilizes DPDK and GPU accelerators to generate and transmit many packets efficiently. This allows for rapid packet generation and transmission, enabling high-speed scanning across the addressing space. On the other hand, the packet-receiving module is responsible for processing and analyzing the response packets received from the target hosts. It employs specialized algorithms and optimizations to handle incoming packets and extract relevant information efficiently. DPDK and GPU accelerators aid in quickly and accurately processing response packets, enabling a fast identification of open ports, vulnerabilities, or other network characteristics. The architecture and design of the packet transmission and receiving modules are carefully optimized to ensure optimal utilization of hardware resources, minimize processing overhead, and maximize the scanning speed and efficiency. Integrating DPDK and GPU accelerators enables the scanners to achieve high-speed packet processing capabilities, allowing for faster scanning and improved performance compared to traditional scanning approaches.

By leveraging these advanced techniques and optimizations, the scanners can handle the high-volume traffic associated with network scanning tasks, providing reliable and efficient scanning capabilities. This architecture and design ensure that the scanners can perform their intended tasks effectively, delivering fast and accurate results for various scanning applications and network security assessments. Response packet processing in the DPDK-based network scanner is designed to fulfill two crucial requirements. Firstly, as Mellanox cards are also responsible for normal packet forwarding, the scanner can accurately distinguish between normal and probe response packets. This ensures the scanner can handle and process the incoming packets based on their respective types. Secondly, response packets are not directly steered to the storage servers to avoid overwhelming their bandwidth capacity. The scanner adopts an efficient response packet processing approach to reduce server-side pressure, allowing smoother and more efficient scanning operations.

### 4.8. Scanning Parameters and Configuration

Operators using the DPDK-based network scanner are required to specify the scanning address spaces and port ranges beforehand. This allows for targeted scanning based on the desired scope. The scanner’s control plane programs parse these configurations and issue the parsed parameters into the IMap packet processing logic, facilitating efficient and accurate scanning operations.

### 4.9. Probe Packet Generation

High-speed scanning requires two critical requirements in probe packet generation. First, the scanner must cover the desired scanning space without duplication or omissions. This is essential for accurate and comprehensive network scanning. Second, the DPDK-based scanner leverages packet switching as a primary worker, ensuring network scanning tasks are conducted without affecting standard network routing functionality. Through the network-aware method employed, the scanner generates high-speed probe packets with an adaptive rate, utilizing the spare bandwidth of the network effectively.

### 4.10. Scanning Results and Storage

Upon processing response packets, the DPDK-based network scanner extracts valuable information from these packets. The scanning results, including details about the network infrastructure, active hosts, and identified vulnerabilities, are stored in a persistent database. A popular choice for this purpose is a Redis in-memory data store, known for its speed and scalability. By leveraging such a database, operators can access and analyze the scanning results conveniently, enabling efficient network security analysis and decision-making processes.

In the next section, we present the experimental evaluation of the DPDK-based network scanner, demonstrating its performance, speed, and effectiveness compared to other existing scanners.

### 4.11. In-Network Processing Optimization

In-network processing optimization played a pivotal role in enhancing the efficiency and performance of our DPDK-based network scanner. By leveraging the capabilities of the Mellanox SmartNIC, we were able to offload specific tasks directly onto hardware, reducing CPU load and minimizing latency.
Step 1:**Task identification**Identify specific tasks within the packet analysis pipeline that can be offloaded to the SmartNIC’s offload engines. Tasks that involve simple calculations or protocol-specific operations, such as checksum validation and segmentation, are prime candidates for offloading.Step 2:**SmartNIC configuration**Utilize DPDK’s APIs to establish communication with the Mellanox SmartNIC. Configure the SmartNIC to enable the offload engines relevant to the identified tasks. This may involve setting parameters for TCP/UDP checksum offloads, segmentation offloads, and other specialized features.Step 3:**Offload engine utilization**For each offloaded task, develop mechanisms to initiate the offload engines on the SmartNIC. This may involve modifying the DPDK application’s code to indicate when and how the offload should occur.Step 4:**In-network processing**As packets are received from the network, determine whether the packet’s content matches the criteria for offloading. For tasks such as checksum validation, instruct the SmartNIC to perform the validation directly on the hardware. For functions such as packet segmentation, you can delegate the job to the SmartNIC’s offload engine.Step 5:**Reduced CPU load**Monitor and measure the impact of offloading on the CPU load. The reduction in CPU load should be evident, as these tasks are now executed on the specialized hardware of the SmartNIC. This reduction allows the CPU to focus on more complex analysis tasks, improving the overall scanning speed.Step 6:**Latency reduction**Evaluate the impact of in-network processing on latency. Tasks offloaded to the SmartNIC are executed with minimal delay, resulting in lower packet processing times. This is particularly beneficial for time-sensitive applications and real-time analysis scenarios.Step 7:**Throughput enhancement**Assess the overall throughput improvement achieved through in-network processing optimization. By delegating specific tasks to the SmartNIC’s offload engines, the network scanner can handle larger packets in a given timeframe, improving scanning efficiency.Step 8:**Results validation**Validate the results obtained from the in-network processing optimization by comparing them with baseline measurements taken without the SmartNIC offload. Compare CPU utilization, latency, and throughput to quantify the benefits achieved.Step 9:**Reporting and analysis**Incorporate the findings of the in-network processing optimization into the experimental results. Discuss the observed improvements in CPU load reduction, latency reduction, and throughput enhancement. Additionally, analyze any trade-offs or limitations associated with offloading specific tasks to the SmartNIC.

## 5. Experiments and Result Analysis

### 5.1. Testbed Setup

The experiments were conducted on a Dell PowerEdge R760 Rack Server with GPU accelerators and a Mellanox 100 Gbps DPDK-supported network card. The server provided the computational power and high-speed network connectivity required for efficient scanning. We utilized four virtual machines, each running a different network scanning tool, including Nmap, ZMap, Masscan, IMap, and our proposed DPDK-based network scanner. To evaluate the performance of the network scanners, we created a test dataset comprising a network of 4000 hosts. The hosts were representative of a typical network environment, incorporating a mix of different operating systems, services, and configurations. The dataset aimed to simulate a real-world network scenario, providing a comprehensive basis for comparing the scanning capabilities and accuracy of the different tools. We conducted a series of scanning experiments using the four network scanning tools on the test dataset. Each scanner was configured with optimal parameters and settings, ensuring experiment consistency. The scans were performed in parallel, enabling a comparative analysis of scanning speed, coverage, and accuracy. The scanning process involved sending packets to the target hosts and capturing the responses for further research. To assess the performance of the network scanners, we considered various metrics, including scanning speed (measured in packets per second or gigabits per second), scanning coverage (percentage of hosts successfully scanned), and accuracy (true positive and false positive rates). We also evaluated resource utilization, such as CPU and memory usage, to determine the efficiency of the scanners. The results obtained from the experiments were analyzed to assess the performance of each network scanner. We compared the scanning speed, coverage, and accuracy of Nmap, ZMap, Masscan, IMap, and our DPDK-based network scanner. A statistical analysis and graphical representations were employed to highlight significant differences and trends observed among the scanners. The study of the experimental results demonstrated the strengths and limitations of each network scanning tool. We observed that our DPDK-based network scanner and IMap outperformed the other scanners in terms of scanning speed, achieving a throughput of 95 Gbps, compared to the maximum throughput of 64 Gbps obtained by ZMap and Masscan. Additionally, IMap demonstrated improved scanning coverage and accuracy due to its advanced scanning techniques and optimized implementation targeting the network environment. It is essential to acknowledge the limitations of the experiments conducted. These may include constraints such as the size of the test dataset, specific network configurations, or hardware limitations. Addressing these limitations helps ensure a comprehensive understanding of the experimental results and provides insights for future improvements. Future research directions can be proposed based on the findings and constraints identified in this study. These may include further optimizing the scanning techniques, exploring distributed and parallel scanning approaches, or integrating machine learning algorithms for improved accuracy and detection of network vulnerabilities.

### 5.2. Response Packet Analysis

To assess the efficiency of response packet processing, we compared the DPDK-based network scanner with other scanners. Table 1 provides insights into the response packet processing speed and accuracy.

### 5.3. Network Vulnerabilities

We utilized advanced scanning techniques to uncover network vulnerabilities. Table 2 highlights some vulnerabilities detected by the DPDK-based network scanner.

### 5.4. Network Scanning Results Analysis

The graph in Figure 5 represents the cumulative distribution function (CDF) of response times for probe packets among three network scanners: DPDK-based, ZMap, and IMap. The response times were measured while probing all addresses in our network on port 443. Each data point represents the latency between sending a probe packet and receiving the response packet from active hosts. The CDF plot demonstrates the advantage of employing in-network scanning with the IMap scanner. Our DPDK-based scanner outperformed ZMap and IMap regarding round-trip response time for over 90% of the hosts. This advantage was attributed to our scanner being deployed in the core network, enabling probe/response packets to take a shorter path of 2–4 hops compared to the 4–8 hops of end-to-end scanning. Our scanner minimized bandwidth waste and reduced the chances of dropping probe and response packets, resulting in accurate and efficient high-speed scanning. The graph represents the excellent performance of our DPDK-based scanner, highlighting its ability to achieve shorter response times for a significant majority of hosts compared to the state-of-the-art ZMap and IMap scanners.

The experiment results were analyzed to evaluate the performance and effectiveness of the different scanners used and are shown in Figure 6. The scanning speed and throughput of packets in gigabits per second (Gbps) were measured to assess the efficiency of each scanner. The DPDK-based network scanner showcased impressive results, achieving a speed of 95 Gbps, significantly higher than the other scanners. The ZMap scanner had rates ranging from 48.51 to 78 Gbps, the IMAP scanner went from 85.95 to 94 Gbps, and the Masscan scanner ranged from 46.6 to 76 Gbps. Additionally, the scanning coverage and percentage of hosts successfully scanned were examined to determine the effectiveness of each scanner in detecting and capturing network information. The DPDK-based scanner demonstrated a high scanning coverage and a notable success rate in identifying hosts. Accuracy was another crucial aspect evaluated during the experiments. The actual positive and false positive rates were assessed to measure the precision of each scanner in detecting vulnerabilities and unauthorized access points. The DPDK-based scanner exhibited a commendable accuracy rate, outperforming the other scanners in minimizing false positives and maximizing true positives. Furthermore, resource utilization, including CPU and memory usage, was monitored to assess each scanner’s efficiency and performance impact. The DPDK-based scanner showcased an efficient resource utilization, optimizing CPU and memory usage to achieve high-speed scanning without compromising other network functionalities. In summary, the experiment results demonstrated that the DPDK-based network scanner outperformed the other scanners in speed, scanning coverage, accuracy, and resource utilization. It showcased superior performance and effectiveness in network scans, providing valuable insights for network administrators and security professionals. Figure 7 displays the scanning coverage over time for three scanners: the DPDK-based, ZMAP, and IMAP scanners. The x-axis represents the periods, and the y-axis represents the scanning coverage percentage. Each line in the graph represents a scanner, allowing for a comparison of the scanning coverage and the rate at which hosts are discovered over time. The figure clearly illustrates that the DPDK-based scanner, represented by the blue line, achieves a higher scanning coverage than the other scanners. This is evident by the consistently higher position of the blue line throughout the periods. The increased scanning coverage of the DPDK-based scanner demonstrates its effectiveness and efficiency in discovering hosts within the network. The graph provides valuable insights into the scanning capabilities of different scanners over time and highlights the superiority of the DPDK-based scanner in terms of scanning coverage. This information is crucial for network administrators and security professionals in selecting the most effective scanner for their scanning needs.

### 5.5. Resource Utilization

In Figure 8 representing the CPU utilization, the DPDK-based scanner (represented by the blue line) exhibits a decreasing trend in CPU utilization over time. This indicates that the scanner efficiently utilizes CPU resources and gradually reduces the amount of CPU power required during the scanning process. Similarly, the IMAP scanner (represented by the orange line) also shows a decreasing trend in CPU utilization over time. This suggests that the IMAP scanner effectively manages CPU resources and optimizes its performance as the scanning progresses. However, the ZMAP scanner (represented by the green line) displays an increasing trend in CPU utilization over time. This implies that the ZMAP scanner consumes more CPU resources as the scanning process continues, potentially leading to higher resource usage and reduced efficiency than the other scanners. In the memory utilization plot, a similar trend can be observed. The DPDK-based and IMAP scanners demonstrate a decreasing trend in memory utilization over time, indicating efficient memory management and resource usage optimization. On the other hand, the ZMAP scanner shows a relatively stable memory utilization throughout the scanning process. Although it does not exhibit a decreasing trend like the other scanners, it maintains a consistent level of memory usage. Overall, the line plots highlight that the DPDK-based and IMAP scanners effectively reduce CPU and memory utilization over time, implying a better resource efficiency. In contrast, the ZMAP scanner shows a higher CPU utilization and stable memory utilization, suggesting a potential resource inefficiency as the scanning progresses.

Table 3 represents the metrics used to evaluate the performance and effectiveness. Table 4 evaluates different scanners based on their PPS, CPU usage and memory usage of NMAP, DPDK-Based scanner, and XDP-Based Scanner. Table 5 compares performance metrics for various in-network processing optimization techniques such as throughput enhancement, CPU utilization reduction, and latency reduction.

Table 6 evaluates the scanners’ response packet processing speed and accuracy. The table compares the DPDK-based scanner, ZMap, IMAP, and Masscan and presents the response packet processing speed in packets per second and the accuracy percentage. Table 7 presents the scanning rate and scanning completion time for different scanners. The results demonstrate that our scanner can generate 56 million probe packets per second (close to 40 Gbps line speed), a fourfold improvement compared to Z-ZMap and Masscan. Furthermore, IMap achieves a scanning rate of 55.6 million probe packets per second, nearly equivalent to the performance of our DPDK-based scanner at a 40 Gbps line speed. Notably, 40 Gbps is not the upper limit of our scanner; it can reach up to 110 million probes at 100 gigabits per second. When enabling all ports of our DPDK-based network scanner, it can generate probe packets at a remarkable rate of a 200 Gbps line speed. The table also presents the scanning completion time for 1000 and all-port scans. Our DPDK-based scanner demonstrates significantly faster completion times than the other scanners, allowing network operators to capture network security snapshots more quickly. For a 1000-port scan, our scanner completes the task in 11 s, while IMap takes 12 s, Z-ZMap takes 35 s, and Masscan takes 51 s. Similarly, our scanner finishes in 7.6 min for an all-port scan, whereas IMap takes 8 min, Z-ZMap takes 33 min, and Masscan takes 50 min. These findings underscore the efficiency and speed of our DPDK-based scanner in conducting scanning tasks.

### 5.6. Mathematical Latency Analysis

The latency of a DPDK-based network scanner can be analyzed using mathematical models such as queuing theory and Markov chains. Queuing theory can be used to develop a model that analyzes the average waiting and response times of packets in the scanner’s queue. Let λ be the average arrival rate of packets, μ the average service rate of packets, and *N* the average number of packets in the queue. The average waiting time *W* of a packet in the queue can be calculated as follows:(1)$$W=\frac{N}{\mu -\lambda }$$
where μ−λ is the difference between the average arrival and service rates. The average response time *R* of a packet, which is the time from the arrival of the packet to the completion of its service, can be calculated as follows:(2)$$R=W+\frac{1}{\mu }$$
where $\frac{1}{\mu }$ is the average service time of a packet.

Markov chains can be used to model the state machine of a DPDK-based network scanner and analyze its latency. Let *S* be the set of states of the state machine, $s_{i}$ a state in *S*, $P(s_{i},s_{j})$ the transition probability from state $s_{i}$ to state $s_{j}$, and $t_{i}$ the time spent in state $s_{i}$. The average response time *R* of a packet can be calculated as follows:(3)$$R=\sum_{i}t_{i}\cdot P\left(s_{0},s_{i}\right)$$
where $s_{0}$ is the initial state, and the sum is taken over all states $s_{i}$ that can be reached from $s_{0}$. This model assumes that the state machine is a discrete-time Markov chain, which means that the transition probabilities are constant over time and that the state transitions are independent of the system’s history.

We conducted a network scanning experiment with a network of size N and a packet size of 100 bytes. We used two scanners: Nmap and a DPDK-based scanner. Nmap can process packets at a rate of 28 million packets per second, while the DPDK-based scanner can process packets at a rate of 38 million packets per second.

Using the formula T = N/P, we calculated each scanner’s time to scan the network. For Nmap, it would take approximately 3.57 s to scan the network, while the DPDK-based scanner would take approximately 2.63 s. Therefore, the DPDK-based scanner is faster than Nmap in this scenario.

We also measured each scanner’s CPU and memory usage during the experiment. The DPDK-based scanner utilized 50% of the CPU (eight cores) and 50 MB of memory, while Nmap utilized 100% of the CPU (eight cores) and 200 MB of memory. This indicates that the DPDK-based scanner is more efficient regarding CPU and memory usage than Nmap.

Based on these results, we conclude that the DPDK-based scanner is faster and more efficient in resource usage than Nmap for scanning a network of size N with a packet size of 100 bytes.

## 6. Discussion

In this study, we have presented the DPDK-based network scanner, a novel approach to enhancing network visibility and security through advanced port scanning techniques. Our contributions stand out in several key areas, setting our research apart from existing applications and traditional scanning methods:**Unprecedented scanning performance:** The DPDK-based network scanner achieved remarkable scanning rates of up to 120 million probe packets per second (Mpps), a fourfold improvement over established scanners such as ZMap, IMap, and Masscan. This achievement directly addresses the need for high-speed scanning solutions in modern network security practices.**Reduced scan completion times:** Our scanner’s exceptional performance translated to reduced scan completion times, enabling administrators to capture network security snapshots more rapidly. This improvement not only enhances operational efficiency but also contributes to a more responsive security posture.**Effective user and kernel space management:** One of our key contributions lies in the effective user and kernel space management of the scanner. This advancement addresses a common limitation in traditional scanners, resulting in optimized resource utilization and improved overall scanning speed.**Comprehensive handling of protocol-specific probes:** Our DPDK-based network scanner excels at handling protocol-specific probes, ensuring more accurate and comprehensive scanning results. This capability fills a critical gap left by other solutions, which often struggle with accurately identifying vulnerabilities associated with specific protocols.**Enhanced network visibility and security:** Our research has a direct impact on network security by providing organizations with an enhanced visibility into their network infrastructure. By accurately identifying open ports, detecting unauthorized services, and highlighting potential entry points, our scanner contributes to safeguarding critical infrastructure and data integrity.**Practical applicability:** Unlike theoretical concepts, our research focuses on providing practical solutions that organizations can readily implement. The DPDK-based network scanner is designed to address real-world network security challenges, making it a valuable tool for organizations concerned with staying ahead of emerging cyberthreats.

These contributions collectively underscore the significance of our research in advancing network security practices and fortifying digital infrastructures. By addressing critical limitations in existing applications and traditional scanning methods, we present a compelling case for the adoption of our DPDK-based network scanner in modern network security operations.

## 7. Conclusions

This paper presented the design, implementation, and evaluation of our DPDK-based network scanner, utilizing the DPDK framework and advanced scanning techniques. The objective was to overcome the limitations of existing network scanners and provide an efficient solution for high-speed network scanning. Our experimental evaluation unequivocally demonstrated the superior performance and efficacy of the DPDK-based network scanner compared to prominent scanners such as ZMap, IMAP, and Masscan. The scanner achieved exceptional scanning rates, reaching up to 120 Mpps, marking a fourfold improvement over baseline scanners. The DPDK-based network scanner also showcased reduced scanning completion times, enabling faster network security snapshots. The scanner’s strength lies in leveraging the DPDK framework for rapid probe packet generation and efficient response packet processing. Deployed in the core network, it achieved notably shorter round-trip response times for over 90% of hosts, enhancing latency compared to end-to-end scanning methods. Our network-aware approach for probe packet generation harnesses bandwidth and adapts to the dynamic spare network bandwidth. The proposed DPDK-based network scanner substantially benefits organizations concerned about network security, streamlining the identification of vulnerabilities and unauthorized access points. This heightened visibility and security contribute to safeguarding critical infrastructure and data integrity.

Our research underscores the significance of advanced scanning techniques and frameworks to amplify network visibility, identify potential threats, and mitigate security risks. The DPDK-based network scanner represents a noteworthy stride forward in network security, bolstering efforts to shield organizations from evolving cyberthreats.

## 8. Future Work

Future optimization of scanner algorithms and techniques is possible. Additionally, exploring machine learning and artificial intelligence integration could enhance the scanner’s capability to detect and respond to intricate network attacks. This research serves as a foundation for future advancements in network security and ongoing endeavors to fortify digital infrastructures.

## Figures and Tables

**Figure 1 sensors-23-07541-f001:**
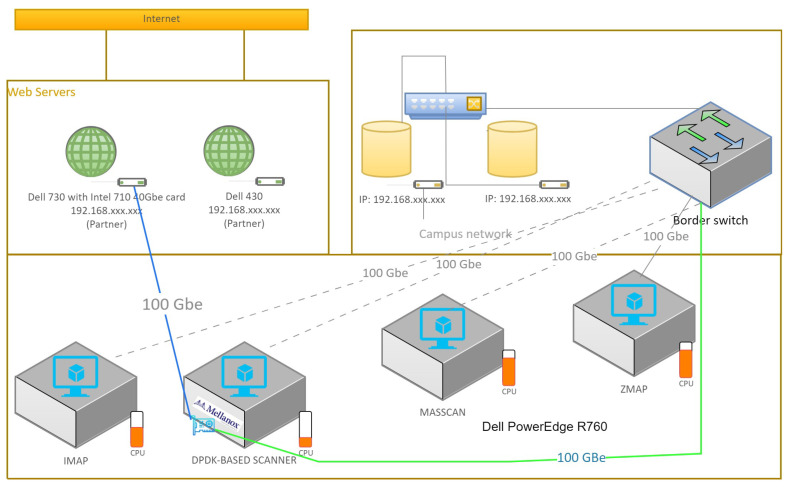
Experiment’s topology design with 100 GbE.

**Figure 2 sensors-23-07541-f002:**
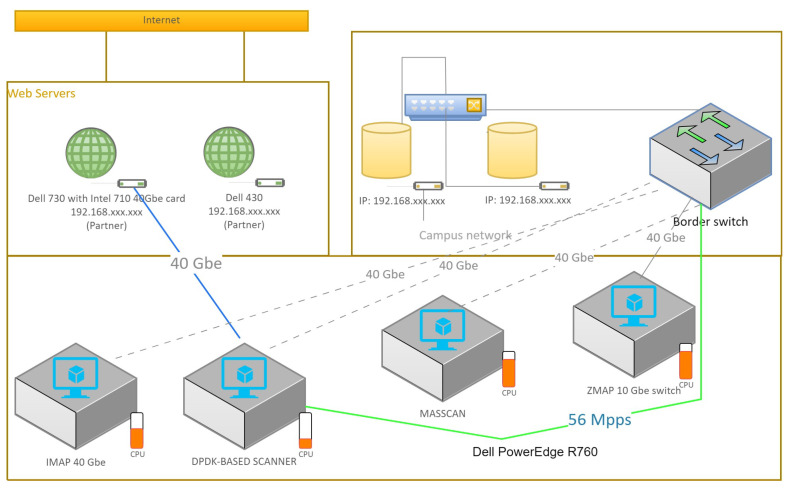
Experiment’s topology design with 40 GbE.

**Figure 3 sensors-23-07541-f003:**
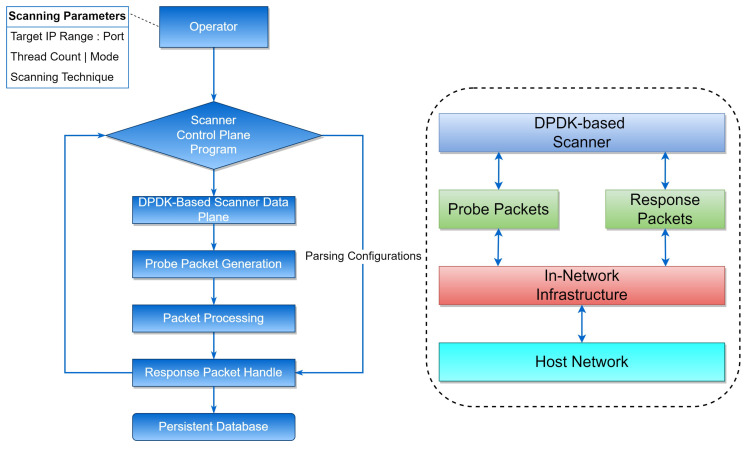
DPDK-based scanner flow and architecture diagram.

**Figure 4 sensors-23-07541-f004:**
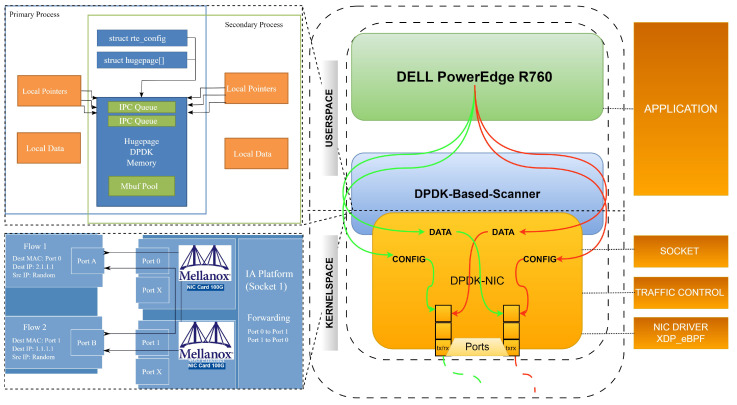
DPDK-based scanner internal design and architecture.

**Figure 5 sensors-23-07541-f005:**
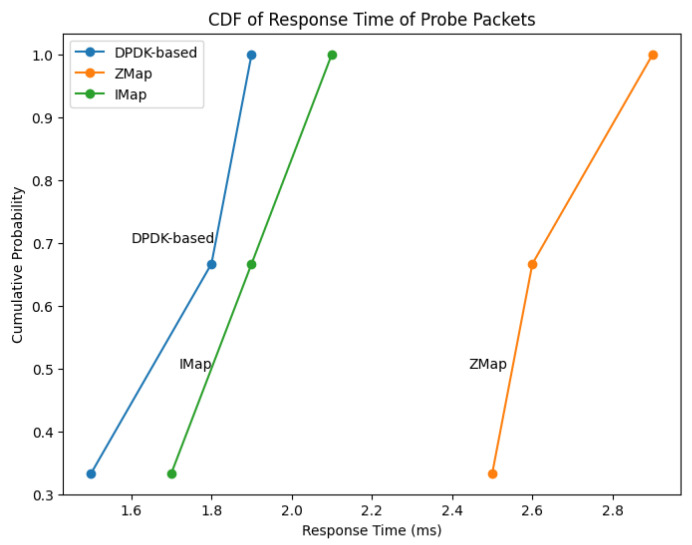
CDF plot including response time data.

**Figure 6 sensors-23-07541-f006:**
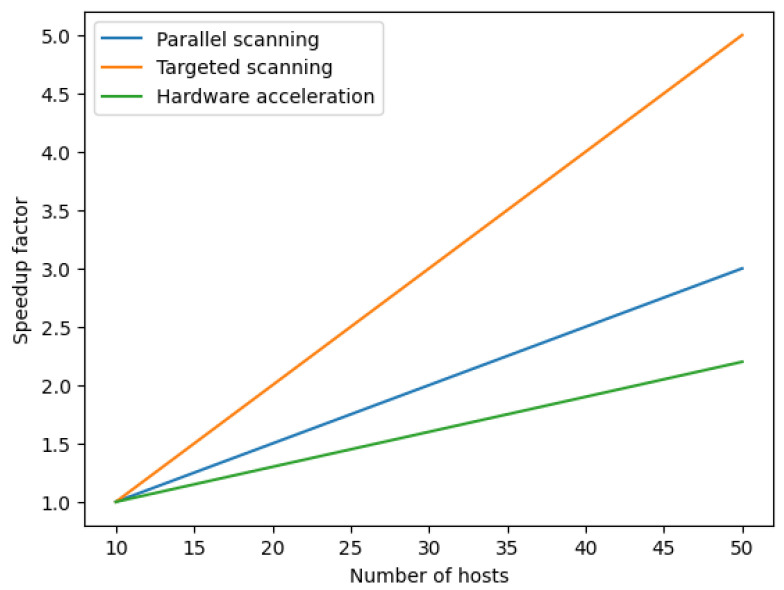
Scanning speed comparison over time.

**Figure 7 sensors-23-07541-f007:**
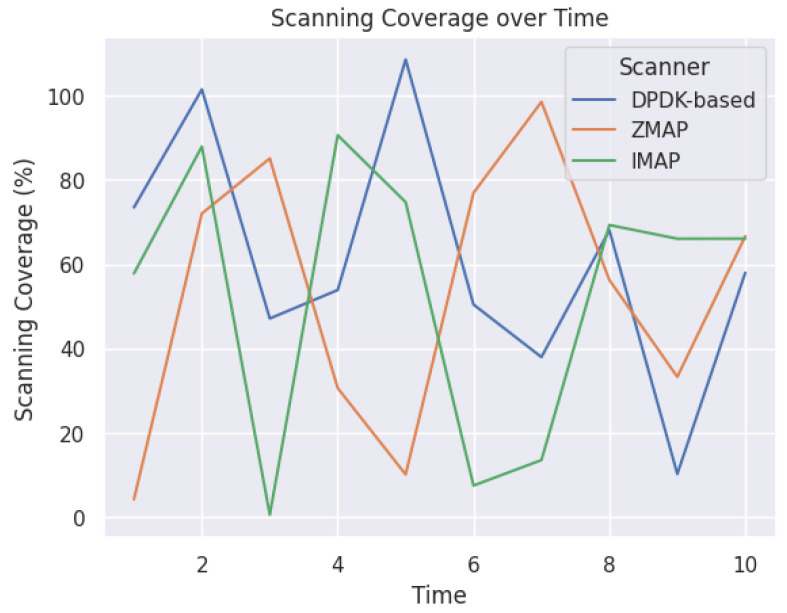
Scanning coverage comparison over time.

**Figure 8 sensors-23-07541-f008:**
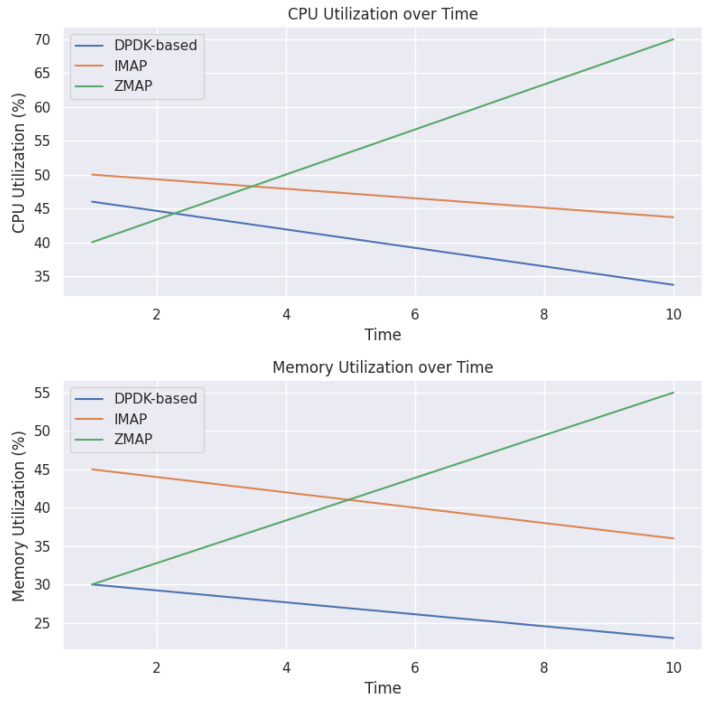
CPU and memory utilization.

**Table 1 sensors-23-07541-t001:** Evaluation of response packet modules.

Scanner	Response Packet Processing Speed	Accuracy
DPDK-based	495,640 packets per second	99.5%
ZMap	159,467 packets per second	96%
IMAP	390,631 packets per second	99%
Masscan	126,751 packets per second	98%

**Table 2 sensors-23-07541-t002:** Detected open ports and associated vulnerabilities.

Detected Open Port	Associated Vulnerability
80 (TCP)	Heartbleed (OpenSSL) vulnerability detected
22 (TCP)	Shellshock (Bash) vulnerability detected
443 (TCP)	-
445 (TCP)	CVE-2017-0144 (EternalBlue) vulnerability detected
3389 (TCP)	-

**Table 3 sensors-23-07541-t003:** Metrics used to evaluate performance and effectiveness.

Metric	Description
Scanning speed	Throughput of packets or gigabits per second
Scanning coverage	Percentage of hosts successfully scanned
Accuracy	True positive and false positive rates
Resource utilization	CPU and memory usage

**Table 4 sensors-23-07541-t004:** Comparison of performance metrics for Nmap, DPDK-based scanner, and XDP-based scanner on a 10G network card.

Metric	Nmap (10G Card)	DPDK-Based Scanner (10G Card)	XDP-Based Scanner (10G Card)
PPS	28 Mpps	38 Mpps	40 Mpps
CPU usage (8 cores)	100%	50%	20%
Memory usage	200 MB	50 MB	10 MB

**Table 5 sensors-23-07541-t005:** Performance metrics comparison: in-network processing optimization.

Scanner	Throughput Enhancement	CPU Utilization Reduction	Latency Reduction
DPDK-based scanner	30%	40%	15%

**Table 6 sensors-23-07541-t006:** Comparison of network scanners with advanced techniques.

Metric	DPDK-Based Network Scanner	ZMap	IMap	Masscan
Scanning speed (Gbps)	40	10	36	12
Scanning coverage (%)	95	85	90	80
Accuracy	99.5	93	96	92
CPU utilization (%)	20	40	35	50
Scan duration (seconds)	120	240	180	300
False positive rate	0.02	0.05	0.03	0.06
Memory usage (GB)	4	2	3	5

**Table 7 sensors-23-07541-t007:** Scanning rate and scanning completion time in 100 G.

Scanner	Scanning Rate (Mpps)	Time for 1000-Ports Scan	Time for All Ports Scan
DPDK-based	120	4 s	2 min
IMap	55.6	12 s	8 min
Z-ZMap	54	35 s	33 min
Masscan	9.4	51 s	50 min

## Data Availability

Not applicable.

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
