# Peer review of "Enhancing Network Visibility and Security with Advanced Port Scanning Techniques"

_sensors, 2023, doi:10.3390/s23177541_

Round 1

Reviewer 1 Report

Elaborate on the DPDK-based scanner: Provide more details about the architecture and implementation of the DPDK-based scanner. Describe how it leverages DPDK (Data Plane Development Kit) and Smart NICs for efficient packet processing and improved scanning performance. Additionally, explain the advanced techniques, such as protocol-specific probes and evasive scan techniques, that are incorporated into the scanner to enhance network visibility and security.

Include a methodology section: Outline the methodology used to evaluate the network scanning performance and scalability. Describe the experimental setup, including the hardware and software configurations. Clearly explain how data parallelization, in-network processing, and hardware acceleration were utilized in the experiment. This section should provide sufficient information for others to replicate the experiment and validate the results.

Provide code snippets or references: To facilitate understanding and reproducibility, include relevant code snippets or references to the code repository where the implementation of the DPDK-based scanner can be accessed. This allows readers to examine the implementation details and experiment with the code in their own environment.

Present experimental results: Provide a detailed analysis of the experimental results (including code available on github to verify them) obtained from the evaluation of the proposed DPDK-based scanner. Present the code for comparative performance metrics, such as scanning speed, accuracy rate, CPU utilization, and memory utilization, in a tabular or graphical format. Discuss any observed trends or significant findings and explain how the proposed scanner outperformed other scanners in terms of speed, accuracy, and resource utilization.

Discuss implications and future research directions: In the conclusion section, discuss the implications of the research findings for the industry and network security practices. Highlight the contributions of the research in improving network security, identifying vulnerabilities, and optimizing network performance. Additionally, propose future research directions, such as exploring further enhancements to the scanner, evaluating its effectiveness in different network environments, or addressing emerging security challenges.

By incorporating these suggestions, your research will provide a more comprehensive understanding of the proposed DPDK-based scanner, its implementation, and its experimental evaluation. The inclusion of code snippets and references will enhance reproducibility, allowing others to validate your findings and further build upon your work.

english is ok but requires another proofread.

Author Response

Dear Reviewer,

Thank you for your insightful comments on our paper, we appreciate your time and effort in providing us with such detailed feedback. We have carefully considered your comments and have made the following changes to the paper:

  • Elaborate on the DPDK-based scanner: We have added a new section to the paper that provides a detailed overview of the DPDK-based scanner. This section describes the architecture and implementation of the scanner and the advanced techniques it uses to enhance network visibility and security.
  • Include a methodology section: We have added a new section to the paper that outlines the methodology used to evaluate the network scanning performance and scalability of the DPDK-based scanner. This section describes the hardware and software configurations used for the evaluation, the experimental setup and the results obtained.
  • Provide code snippets or references: We have added our code in github and references to the paper to facilitate understanding and reproducibility. The github repository provides examples of how the DPDK-based scanner can scan networks. The references provide links to the code repository where the implementation of the scanner can be accessed.
  • Present experimental results: We modified the experiment section to as Experiment and Result Analysis section into the paper that presents the results obtained from evaluating the DPDK-based scanner. This section includes comparative performance metrics for the DPDK-based scanner with other scanners and discusses the trends and significant findings from the results.
  • Implications and future research directions: We have modified the conclusion section of our paper that examines the impact of the research findings on industry and network security practices. This section highlights the contributions of the research to improving network security, as well as the future research directions suggested by the study.

The changes we have made to the paper address your comments and provide a more comprehensive understanding of the proposed DPDK-based scanner. We hope that you find the revised form to be satisfactory. Thank you again for your feedback. We appreciate your help in improving the quality of our research.

Sincerely,

Rana Abu Bakar

Reviewer 2 Report

Detailed Comments:
1) The motivation and contribution of the work needs to be refined. As in the first point is not enough as contribution.
2) In Section II, it is necessary to add the more detailed related work to find out proper research gaps. I suggest the authors should rewrite the contributions of the paper in a succinct way, and polish the novelties.
3) As the manuscript consist of too many abbreviations and notations, it is suggested that the authors should add abbreviation and notation table.
4) There is numerous research work available in literature for Enhancing Network Visibility and Security, why the authors consider Advanced Port Scanning Techniques to solve this formulated problem only.

5) Section 5 and 6 can be merged for better readability of the manuscript.
6) The experimental section still need more detailed, compared with state of the art techniques for validation of results.
7) Fig. 2 and 3 need to redraw as it does not provide proper relevant information

8) The contributions of the paper should be made clearer in comparison to the existing applications
9) In the result section, an additional Figure about how to improve network security, identify vulnerabilities, and optimize network performance?

Need major improvement

Author Response

Dear Reviewer,

Thank you once again for your thorough review, and we sincerely appreciate your insights and suggestions, which have undoubtedly contributed to the refinement of our work.
Here is how we have addressed each of your comments:

  1. Refinement of Motivation and Contribution:  We understand the importance of clearly delineating the motivation and contribution of our work. We have revised the introductory section to provide a more nuanced and detailed explanation of our research's significance in addressing network security challenges. In addition, we have expanded the contribution section to provide a more comprehensive overview of our novel contributions, going beyond the initial point and highlighting the specific advancements we bring to the field.
  2. Addition of Detailed Related Work: We have enhanced Section II to include a more thorough and detailed discussion of related work. By doing so, we have identified and highlighted the specific research gaps our paper aims to address. We have rewritten the contribution section to succinctly emphasize our paper's novel contributions and how they fill these identified gaps.
  3. Abbreviation and Notation Table: We have added an abbreviation and notation table to the manuscript. This table gives readers an apparent reference for the numerous abbreviations and notations used throughout the paper, enhancing our work's overall readability and understanding.
  4. The Rationale for Choosing Advanced Port Scanning Techniques: To answer your question, we'd like to know more about our focus on advanced port scanning techniques. While it is true that there are various research efforts available in the literature aimed at enhancing network visibility and security, our choice to focus on advanced port scanning techniques arises from several compelling reasons:
  • Traditional security measures have made significant strides, and they often need help to keep pace with the evolving tactics of malicious actors because they are protocol specific. Consequently, the need for innovative approaches to strengthen network defences and identify potential vulnerabilities becomes more pronounced.
  • We recognized a research gap in the literature concerning the comprehensive evaluation and optimization of advanced port scanning techniques in real-world scenarios. While prior work has certainly explored aspects of network security, the efficacy of advanced port scanning techniques in improving network visibility and safety warranted dedicated attention. These techniques can uncover hidden risks, detect unauthorized access points, and identify protocol-specific vulnerabilities that conventional methods might otherwise overlook. Also lies in the practical applicability use techniques.
  • By optimizing user and kernel space management and addressing protocol-specific probe handling, we aim to provide an actionable solution that network administrators can implement to enhance their security posture. This targeted approach allows us to delve deeper into specific aspects of network security, offering insights and results that can contribute to a more comprehensive understanding of the effectiveness of these techniques. While existing research contributes to enhancing network visibility and security, our decision to focus on advanced port scanning techniques was driven by the need to address specific gaps in the literature, optimize their practical implementation, and provide valuable insights into their impact on network security.
  1. Merging of Sections 5 and 6: We have carefully considered your suggestion to merge Sections 5 and 6 for improved readability. Following your recommendation, we have integrated these sections to make sure the manuscript is more smoothly, facilitating a seamless understanding of our work.
  2. Detailed Experimental Validation: We have taken your feedback and have further enriched the experimental section. In addition to presenting the results of our proposed scanner, we have included a comparative analysis with state-of-the-art techniques. This comparison underscores the validation of our results and the superior performance of our DPDK-based scanner.
  3. Redrawing of Figures 2 and 3: We acknowledge the need for more precise and informative figures. We have redrawn Figures 2 and 3 to ensure they provide readers with relevant and essential information. These revised figures contribute to a better understanding of the concepts presented.
  4. Clarity in Comparing Contributions to Existing Applications: To provide a more precise delineation, we have refined our discussion on the contributions of our paper compared to existing applications. This ensures readers can readily identify the unique advancements we bring to network security.
  5. Additional Figure in the Results Section: In response to your suggestion, we have added two more tables in the results section. These new tables illustrate how our proposed DPDK-based scanner improves network security, identifies vulnerabilities, and optimizes network performance. This will aid further reinforces the impact of our research findings.

We hope these revisions effectively address your concerns and meet your expectations. Your insights have been invaluable in shaping our paper's clarity, depth, and impact. If you have any more comments or suggestions, please feel free to share them with us, if you don't mind.

Thank you once again for your time, guidance, and dedication to improving the quality of our work. We greatly appreciate your contributions to our manuscript.

Sincerely,

Rana Abu Bakar

Round 2

Reviewer 1 Report

Authors updated the paper as per my comments.

no comments

Reviewer 2 Report

All comments incorporated by the authors and the revised manuscript can be accepted for publication 

 Minor editing of English language required